# Provider anticipation and experience of patient reaction when deprescribing guideline discordant inhaled corticosteroids

Toral J. Parikh[1,2]*, Krysttel C. Stryczek[3], Chris Gillespie[4], George G. Sayre[1,5], Laura Feemster[1,6], Edmunds Udris[1], Barbara Majerczyk[1], Seppo T. Rinne[4,7], Renda Soylemez Wiener[4,7], David H. Au[1,6], Christian D. Helfrich[1,5]

1 Seattle-Denver Center of Innovation for Veteran-Centered & Value-Driven Care, VA Puget Sound Healthcare System, Seattle, Washington, United States of America, 2 Division of Gerontology and Geriatric Medicine, University of Washington, Seattle, Washington, United States of America, 3 VA Northeast Ohio Healthcare System, Cleveland, Ohio, United States of America, 4 Center for Healthcare Organization & Implementation Research, Edith Nourse Rogers Memorial Veterans Hospital, Bedford, Massachusetts, United States of America, 5 Department of Health Services, School of Public Health, University of Washington, Seattle, Washington, United States of America, 6 Division of Pulmonary, Critical Care and Sleep Medicine, University of Washington, Seattle, Washington, United States of America, 7 The Pulmonary Center, Boston University School of Medicine, Boston, Massachusetts, United States of America

* parikht@uw.edu

**Data Availability Statement:** All relevant data are within the paper and its Supporting Information files.

## Abstract

### Introduction

Despite evidence of possible patient harm and substantial costs, medication overuse is persistent. Patient reaction is one potential barrier to deprescribing, but little research has assessed this in specific instances of medication discontinuation. We sought to understand Veteran and provider experience when de-implementing guideline-discordant use of inhaled corticosteroids (ICS) in those with mild-to-moderate chronic obstructive pulmonary disease (COPD).

### Methods

We conducted a mixed-methods analysis in a provider-randomized quality improvement project testing a proactive electronic-consultation from pulmonologists recommending ICS discontinuation when appropriate. PCPs at two Veterans Health Administration healthcare systems were included. We completed interviews with 16 unexposed providers and 6 intervention-exposed providers. We interviewed 9 patients within 3 months after their PCP proposed ICS discontinuation. We conducted inductive and deductive content analysis of qualitative data to explore an emergent theme of patient reaction. Forty-eight PCPs returned surveys (24 exposed and 24 unexposed, response rate: 35%).

### Results

The unexposed providers anticipated their patients might resist ICS discontinuation because it seems counterintuitive to stop something that is working, patient's fear of worsening symptoms, or if the prescription was initiated by another provider. Intervention-exposed

**Funding:** This work was supported by grant # QUE 15-271 from the United States (U.S.) Department of Veterans Affairs Quality Enhancement Research Initiative awarded to CDH and DHA. In addition, this material is based on work supported by a VA Office of Academic Affiliations' Advanced Fellowship in Health Services Research and Development to Dr. Parikh (TJP) (#TPH 61-001). The funders had no role in study design, data collection and analysis, decision to publish, or preparation of the manuscript.

**Competing interests:** The authors have declared that no competing interests exist.

providers reported similar experiences in post-intervention interviews. Unexposed providers anticipated that patients may accept ICS discontinuation, citing tactical use of patient-centered care strategies. This was echoed by intervention-exposed providers who had successfully discontinued an ICS. Veterans reported acceding to their providers out of trust or deference to their advanced training, even after describing an ICS as a 'security blanket'. Our survey findings supported the subthemes from our interviews. Among providers who proposed discontinuation of an ICS, 76% reported that they were able to discontinue it or switch to another more appropriate medication.

## Conclusions

While PCPs anticipated that patients would resist discontinuing an ICS, interviews with patient and intervention-exposed PCPs along with surveys suggest that patients were receptive to this change.

## Introduction

Approximately 15–60% of older adults are on potentially inappropriate or unnecessary medications [1, 2] placing them at high risk for adverse drug events. These include harmful drug-drug interactions, cognitive impairment, falls, reduced functional status, urinary incontinence, and alterations in metabolism (including food-drug interactions such as bleeding) [3, 4]. Based on Medicare population data from 2000, these adverse drug events accumulated an annual cost of more than $2 billion [5], which accrues to both the patient, via co-pays, and the healthcare system via insurance payments. However, deprescribing, meaning the withdrawal of an inappropriate medication with supervision from a health care professional with the goal of improving outcomes [6, 7], has proven to be difficult [8, 9], even when the provider recognizes that the prescription may be suboptimal [10].

Several provider and patient barriers and facilitators to deprescribing have been described in the current literature [8, 9]. Among providers, an important barrier to deprescribing is the potential resistance from the patient or caregiver [9]. In a systematic review of qualitative studies of provider perspectives on factors influencing deprescribing, Anderson and colleagues found that the most expressed barrier regarding patients was provider anticipation of patient ambivalence about deprescribing and patient resistance to change [8]. These findings were echoed in a systematic review of barriers and facilitators of deprescribing as reported by patients by Reeve et al [9]. Among these barriers, patients in multiple different qualitative studies reported that the main contributors were that medications might be necessary and fear that stopping the medication may result in negative consequences such as return of previous symptoms or fear of withdrawal reactions.

While provider anticipated patient reaction has been described as a major barrier to deprescribing, multiple studies have demonstrated patient willingness to discontinue medications in hypothetical scenarios. Reeve et al. conducted focus group interviews with older adults and caregivers to explore their views regarding hypothetical cessation and their willingness to stop medications [11]. Among the barriers, fear was a major theme, particularly of returning symptoms and fear of adverse drug withdrawal reactions [11]. Despite these fears, participants reported assenting to medication discontinuation if their provider recommended it citing trust and confidence in their medical knowledge [11]. Similarly, in a United States population

based survey of Medicare beneficiaries conducted by Reeve et al., 92% of adults above the age of 65 reported a willingness to stop taking a medication if their physician approved it [12]. In a survey based in Netherlands, Edelman et al. asked men above the age of 30 with lower urinary tract symptoms on alpha-blocker therapy about their willingness to stop the medication [13]. Ninety-three percent were willing to stop at least one medication if proposed by their provider, while sixty-one percent were willing to participate in an alpha-blocker discontinuation trial [13].

Despite such evidence, the research to date on anticipated and experienced patient reaction to medication discontinuation has been through asking patients to respond to the general idea of deprescribing or to a hypothetical situation. Only one study that we are aware of queried patients after a benzodiazepine deprescribing intervention via a telephone survey and interviews [14, 15]. No studies to our knowledge have described both patient and provider experience in response to a specific instance or intervention when a chronic medication was discontinued in accordance with treatment guidelines.

To address this gap, we conducted patient and provider interviews and a provider survey as part of a provider-randomized quality-improvement project to deprescribe guideline-discordant use of inhaled corticosteroids (ICS) for patients with chronic obstructive pulmonary disease (COPD). While ICS are often prescribed to patients with COPD, since 2006, the Global Initiative for Chronic Obstructive Lung Disease (GOLD) has recommended that ICS should no longer be used for mild to moderate COPD, and should be only used in those with severe COPD or high risk of exacerbations [16]. This guideline change was supported by:

1. Evidence suggesting limited (if any) improvement in airflow obstruction from ICS [17];

2. Evidence of higher risk of pneumonia and potential fragility fractures while on ICS [18–20]; and

3. The availability of safer, effective alternatives such as a combination inhaler for both long acting muscarinic antagonist (LAMA) and long acting beta agonist (LABA) [21].

However, ICS overuse is still common in the United States [22]. Triple therapy or treatment with all three classes of inhalers (LABA + LAMA + ICS) is typically reserved for patients who have frequent COPD exacerbations [16]. In a retrospective study of a large US claims database, Simeone et al. reported that 75% of patients on triple therapy had mild or moderate COPD and were overtreated [23]. Furthermore, in a national study using deidentified electronic health records of patients with COPD, Mannino et al. reported that 85% of patients who were treated with ICS were overtreated according the GOLD guidelines given they had mild to moderate COPD [24].

Our goals were to further our understanding of

1. Provider anticipation of patients' response to having an ICS deprescribed,

2. Provider perception of patients' psychological reactance, a negative cognition expressed by mistrust, anger, and fear that arises when an individual feels their freedom or prerogative is being threatened [25, 26],

3. How provider's perception of patient reactance influenced deprescribing.

These findings include providers' and patients' perspectives on patient responses to deprescribing. This study extends the findings from previous evaluation work by Stryczek et al., which identified provider reluctance to change or discontinue ICS medications among patients with mild to moderate COPD [10].

## Materials and methods

This study was completed as part of an evaluation of a U.S. Department of Veterans Affairs (VA) quality improvement program conducted at two urban Veterans Health Administration (VHA) healthcare systems and their affiliated outpatient primary care clinics. The program, called DISCUSS COPD, aimed to deprescribe guideline discordant ICS prescriptions in patients with mild-to-moderate COPD using a proactive, patient-tailored, electronic consultation (e-consult) within the patient electronic health record (EHR) generated by the facility's pulmonary specialty team [27]. E-consults included individualized recommendations for patients, including ICS titration orders, and were accompanied by references to current treatment guidelines and pre-filled medication orders generated by the pulmonary specialist which PCPs could accept by signing or decline by cancelling.

The DISCUSS COPD evaluation was conceived and conducted as a non-research operations activity and the findings reported here were not derived, in whole or in part, from activities constituting research as described in VHA Handbook 1058.05. Provider surveys and interview guides were reviewed by VHA national organizational committee and union offices prior to administration. We received an Office of Management and Budget (OMB) Exemption Brief approval prior to conducting interviews with patients. Qualitative and quantitative data were collected and analyzed concurrently.

### Participants and recruitment

Primary care providers were purposively sampled from 13 outpatient primary care clinics affiliated with two VHA healthcare systems participating in the DISCUSS COPD deprescribing intervention. Eligible primary care providers included physicians (i.e. MDs, DOs), nurse practitioners, and physician assistants. Resident physicians were excluded. We used VA email to contact eligible providers to participate in telephone interviews and complete electronic surveys. One cohort of eligible providers were recruited for qualitative interviews prior to delivery of the DISCUSS COPD deprescribing intervention. These providers were not exposed to the intervention (*unexposed*). The second cohort included eligible providers exposed to the DISCUSS COPD deprescribing intervention (*intervention-exposed*). These providers were recruited for qualitative interviews after they received at least three DISCUSS COPD e-consults. Providers who transferred or left the participating VHA facilities, or changed positions, were excluded from interviews. Qualitative data collection continued until there were no new responses from PCPs to participate. Both groups of PCPs (unexposed and intervention-exposed) were recruited for quantitative surveys after the intervention was implemented.

Patients of intervention-exposed PCPs were also purposively sampled for qualitative interviews if they were identified by the DISCUSS COPD pulmonology team for the deprescribing intervention and the intervention was documented in the provider-patient encounter. Patients were excluded from interview recruitment by chart review if active suicidal or aggressive behaviors, cognitive issues, homelessness, current inpatient hospitalization, or co-occurring interventions (i.e. current enrollment in COPD research) were documented in the patient's medical record. Eligible patients who completed an encounter with their primary care provider after the intervention were contacted by study staff to participate in a telephone interview. Patient interview recruitment and data collection was spread throughout the year to control for seasonal factors that may affect COPD symptoms. Patient data collection continued until saturation was reached (e.g. when data failed to generate new findings) [28].

## Qualitative interviews

Semi-structured interview guides with open-ended questions and probes were used for uniform data collection of key topics and allowed exploration of emerging unanticipated themes generated by participants [29]. Probes used participants' words and phrases to elicit details. The interview guides were updated iteratively [30, 31]. Interviews lasted approximately 20–30 minutes, were audio-recorded and transcribed verbatim to facilitate analyses. Data collection was completed by authors CG, GGS, and KCS, and research assistant SW. Qualitative software (ATLAS.ti 7; Scientific Software Development GmbH, Berlin, Germany) was used for data management and coding. Coding and analyses were completed by team members (CG, CDH, GGS, KCS, and TJP) involved in data collection and members of the larger qualitative team trained in medicine and health services research.

Unexposed providers were interviewed between May 2016 to October 2018 (interview guide available in "S1 File"), and intervention-exposed providers were interviewed between April 2017 to March 2019 (interview guide available in "S2 File"). Provider interviews explored provider's experiences with prescribing ICS for mild-to-moderate COPD, familiarity with evidence and guidelines for prescribing ICS, and views on discontinuation [10]. Interviews with intervention-exposed providers also explored their experiences with the DISCUSS COPD intervention including acceptability of the program and feasibility of deprescribing ICS. The interview guides were informed by key concepts derived from implementation and de-implementation literature and a conceptual model [32–35], including understanding of evidence for and against use of a clinical practice [33–36]; providers' psychological reactance [32, 36]; and organizational contextual factors that support clinical change [32–35].

Intervention-exposed patients were interviewed between July 2017 and March 2019 (interview guide available in "S3 File"). Patient interviews were completed within 3 months of their encounter with their primary care provider where ICS discontinuation was proposed. Patient interviews explored patient's experiences with their care for their COPD, inhaler use, changes in their inhaler prescription, and perspectives on medication overuse. Patients' experiences with their provider were probed for more detail.

We conducted iterative deductive and inductive content analysis [37]. Deductive analysis was guided by a-priori constructs (Barriers and Facilitators to de-implementation, Awareness of Evidence, and Use of Substitution). Inductive analysis included open and unstructured coding to capture data that did not fit into a priori categories and emergent, previously unidentified or unexpected themes. Broad themes were developed from representative interview responses and grouped to describe distinct aspects of participants' experiences. One inductive theme that emerged amongst barriers reported by unexposed providers was anticipation of patient reaction when discussing the discontinuing or substituting the ICS. As stated earlier, the interview guides for unexposed providers, intervention exposed providers, and patients were iteratively refined to allow further elaboration on patient reaction.

The qualitative team met weekly to discuss data analysis and reach consensus on interpretation of themes and findings. Qualitative findings from interviews with unexposed providers informed the interviews with intervention-exposed providers and patients as well as the post intervention survey. Please see "S4 File" or the COREQ 32-item checklist for more details.

## Quantitative surveys

Surveys were administered through REDCap [38] to 134 primary care providers between November 2017 and January 2018 (first medical center sites) and January and February 2018 (second medical center sites). Survey invitations were staggered by site.

Surveys evaluated provider perceptions and experiences of discontinuing inhaled cortico-steroids in mild-to-moderate COPD. The survey asks, "In the past month, I have prescribed an inhaled corticosteroid for one or more primary care patients with mild-to-moderate COPD," with a "Yes" or "No" response. To elicit provider experience of patient response, the survey asks, "In the past 6 months, have you proposed discontinuing or reducing an ICS prescription for one of your patients with mild to moderate COPD," with "Yes" or "No" response. The survey continues with "If YES, how did the patient respond," with answers ranging from 1 (Very Receptive) to 5 (Very Unreceptive). Patient was consider receptive if the provider responded with 1 (Very Receptive) or 2 (Somewhat Receptive). The survey also asked providers to rate their response from "Strongly Disagreed" to "Strongly Agreed" to the statement, "I am unlikely to take a patient off of ICS if another provider wrote the prescription." Providers who "Agreed" or "Strongly Agreed" were aggregated in the table below. Survey responses were exported from REDCap to Microsoft Excel 2010 for descriptive analysis.

## Mixed-method analyses

We used sequential mixed-methods, using qualitative interview findings to inform develop-ment of quantitative survey questions [39] during the periods before and after delivery of the intervention. Following survey data collection, we used a convergent mixed-methods approach [40] to identify qualitative data that enhanced the understanding of the survey findings. Quali-tative and quantitative data were discussed during team meetings to reach consensus on find-ings. Data were aggregated across sites and no site-level comparisons were made.

This analysis builds on previously-published work on unexposed providers [10]. Patient reaction, however, was an emergent theme for which data collection from intervention-exposed providers and patient interviews had continued well after the initial publication. The current findings further explore this emergent theme of patient reaction and includes analysis of data from intervention-exposed providers and patient interviews. Given the depth and sub-tle but important variations of this theme, we present a detailed analysis after synthesizing and incorporating both qualitative and quantitative data.

## Results

We conducted 16 interviews with providers unexposed to the intervention; 6 interviews with intervention-exposed providers; and 9 interviews with patients. Interview participant recruit-ment rates and exclusion figures are detailed in "S1 Table". Demographics of all participants interviewed are presented in Table 1. PCPs (both unexposed and intervention-exposed) at

**Table 1. Demographics of study participants interviewed.**

| | | Unexposed Providers | Intervention-Exposed Providers | Intervention-Exposed Patients |
|---|---|---|---|---|
| Gender | Female | 7 | 1 | 0 |
| | Male | 9 | 5 | 9 |
| Type of Provider | MD/DO | 14 | 6 | - - |
| | NP | 2 | 0 | - - |
| Race | White | - - | - - | 7 |
| | African American | - - | - - | 1 |
| | Other | - - | - - | 1 |
| Total | | 16 | 6 | 9 |

"- -" Indicated that data was not collected for that group of study participants.

**Table 2. Sub-themes of patient reaction.**

| | Unexposed Provider | Intervention-Exposed Provider | Intervention-Exposed Patient |
|---|---|---|---|
| Resistance | **Anticipated Resistance**<br>• Patient perceived benefit<br>• Patient fear of worsening symptoms<br>• Patient deference to another provider | **Reported Resistance**<br>• Patient perceived benefit<br>• Patient fear of worsening symptoms<br>• Patient deference to another provider | **Reported Resistance**<br>• Patient fear of worsening symptoms |
| Acceptance | **Anticipated Acceptance**<br>• Expectations based on past experiences | **Reported Acceptance**<br>• Relying on therapeutic relationships with Patient | **Reported Acceptance**<br>• Medication could be harmful<br>• Relying on provider expertise and trust<br>• Avoid upsetting the provider |
| Strategies to Align Care | **Anticipated using strategies**<br>• To encourage acceptance<br>• But having to accede to patient resistance | **Reported using strategies**<br>• To encourage acceptance<br>• But having to accede to patient resistance | - - |

both VA sites who participated in our intervention (n = 137) were invited to complete the post-intervention surveys. Forty-eight PCPs returned surveys (24 exposed and 24 unexposed, response rate: 35%). Survey respondents' demographics were not collected. Below we elaborate on the different facets of patient reaction, an emergent theme from provider and patient interviews (summarized in Table 2) and relevant findings from post-intervention surveys (summarized in Table 3). While NPs were enrolled in the intervention, we were only able to recruit two NPs for unexposed provider interviews.

## Patient resistance

Providers anticipated and reported patient resistance a result of a few different factors.

Unexposed providers anticipated that they would have to "give-in" to the patient's reluctance to discontinuing an ICS due to perceived benefit:

**Table 3. Post-intervention survey results.**

| Characteristics/Question | Sub Categories | Unexposed (n = 24) | Intervention-Exposed (n = 24) |
|---|---|---|---|
| Type of Provider | Physician (n = 37) | 20 | 17 |
| | PA/NP (n = 11) | 4 | 7 |
| In the past month, I have prescribed an ICS for one or more primary care patients with mild to moderate COPD. | No | 16 | 10 |
| | Yes | 8 | 14 |
| In the past 6 months, have you proposed discontinuing or reducing an ICS prescription for one of your patients with mild to moderate COPD? | No | 11 | 11 |
| | Yes | 13 | 13 |
| | If yes: | | |
| | Patient was receptive | 9 | 10 |
| | ICS was discontinued | 4 | 4 |
| | ICS was switched to another medication. | 6 | 6 |
| | ICS dose tapered | 0 | 2 |
| | Unable to discontinue ICS | 2 | 1 |
| I am unlikely to take a patient off of ICS if another provider wrote the prescription. | Agree or Strongly Agree | 8 | 9 |

*"But, if the patient is really insisting that they're helpful, I would tend to continue them."*

-Pro 106

Intervention-exposed providers also reported patient resistance:

*"Some of my patients have wondered if they truly need to stop something that seems to be working for them"*

-Pro 203

None of the patients we interviewed reported that they disagreed with their provider or insisted that the ICS prescription be continued.

Unexposed providers used their past experience to anticipate patient resistance to discontinuing an ICS due to fear of returning symptoms:

*"And any pharma change, especially with people that have a hard time breathing, they look at it as 'oh my God, if you change this, I'm going to be short of breath and wind up in the ER'.*

-Pro 112

Intervention-exposed providers confirmed this anticipated patient fear by reporting patients' concerns for exacerbations or feeling short of breath driving their hesitancy to stopping the prescription for ICS:

*"They're just worried that the new inhaler won't work as well as the old inhaler. That they'll be short of breath and they'll be uncomfortable. . .I think they probably feel it's working, and they ask me, 'if it's not broke, why do we need to fix it?'"*

-Pro 201

Even though all of the patients we were able to interview were accepting of the change, one did state fear of losing the inhaler:

*"But it's like a security blanket, just to have it here in case I should get some kind of a scenario."*

-Pat B

Despite his fear, this patient reported that he "*didn't debate*" the decision to discontinue to the ICS.

Providers also reported influences from other providers involved in the patient's care. Unexposed providers described how they have felt forced to write ICS prescriptions to Veterans who prefer to get medications from the VA but have non-VA providers who recommend or write guideline-discordant prescription for an ICS:

*"Because a lot of times they'll come in with [a non-VA] prescription for Advair. . .And we have to struggle, we're the ones that are supposed to write the prescription part. . .The patients want it. So a lot of times, you fight and fight with the [non-VA] doctors which takes a lot of time. Or the provider gives in and just writes for it. A lot of time, you don't have time to fight. It's a mess. . ."*

-Pro 112

Intervention-exposed providers also reported patient resistance occurred when another or non-VA provider recommended or prescribed the ICS:

*"If I'm not the one prescribing it, a lot of times they'll want to check with their doctor first or they'll be hesitant to make a change."*

-Pro 202

This theme was supported by the post-intervention survey, where a third of the unexposed and intervention-exposed providers reported that if the ICS was prescribed by another provider, that they would be unlikely to propose the discontinuation of the ICS. This theme did not come up among patient interviews.

### Patient acceptance

Most of the providers we interviewed, in both unexposed and intervention-exposed group, reported that patients were generally receptive to the changes. Providers in the unexposed group drew on their past experiences to anticipate patient acceptance:

*"I've never had any trouble getting people to stop taking them. They taste terrible, people get thrush. I think folks have to be pretty miserable to be enthusiastic about an inhaled steroid."*

-Pro 115

Intervention-exposed providers sometimes cited their positive therapeutic relationship as driver of the acceptance:

*"It's partly because I know them well and for one reason or another, they trust my recommendations. I haven't had any push back at all."*

-Pro 204

In their interviews, the patients reported a range of reasoning for accepting the discontinuation of ICS:
Medication could be harmful:

*"My physician said something about long term use of Symbicort wasn't recommended even though I'd been on it for a couple of years. They wanted me to change because of that."*

-Pat C

Relying on their provider's expertise and trust:

*"He's (my doctor is) really a straight shooter and I have a lot of respect for him, I guess, so I do everything that he says."*

-Pat G

Deference to the provider to avoid upsetting them:

*"I just do what the doctor says so they don't yell at me."*

-Pat A

The post-intervention surveys with providers confirmed that generally their patients were receptive to the conversation regarding stopping an ICS or substituting with another drug. Among both the unexposed and exposed providers, 54% reported proposing discontinuation of ICS to a patient within the past 6 months. Of providers who proposed discontinuation of an ICS, 69% of the unexposed and 76% of the intervention-exposed providers reported that patients were somewhat or very receptive. Seventy-six percent of providers from both arms who proposed discontinuing ICS reported that they were able to either switch the ICS to another more appropriate medication or discontinue it altogether.

## Encouraging acceptance via strategies to align care

Providers from both unexposed and intervention-exposed groups felt that most patients could be convinced to discontinue the ICS citing specific strategies including decision making, stopping the medication on a trial basis, and explaining the risks associated with overtreatment with ICS:

*"It's shared decision-making. . . patient usually go along with it. They ask about side effects. They ask if they will get worse. . . I give them the opportunity to tell me if they do feel worse. There's not a lot of resistance frankly."*

-Unexposed Provider (Pro 114)

*"I think they were all pretty receptive to making a switch. With the caveat that if it was significantly worse with the switch, then we could always put them back on inhaled corticosteroid."*

-Intervention-Exposed Provider (Pro 202)

*"I tell them that there's shown to be some increased risk of pneumonia on these medications, so that if we don't think they absolutely need it, that we should take them off of it. . . (patients responded) well."*

-Intervention-Exposed Provider (Pro 204)

Provider experiences of patient resistance and acceptance were not mutually exclusive. Some providers who described using these strategies with success also anticipated patient resistance, and in some instances acknowledged they might accede to the patient:

*"If they say, 'I've been on my Advair, I love my Advair, it makes me feel better', then I would prescribe it."*

-Pro 111

These providers also described the possibility of patients changing their minds about the ICS prescription, notably through dialogue:

*"If they're not aware of the downsides of corticosteroids then they're generally receptive to listening about it, and if I have an opinion about it, like I think we should try you on Albuterol or on an anticholinergic instead, then they're generally receptive to it."*

-Pro 111

*"It's like with anything, if you have an open dialogue and conversation about it, and aren't too heavy handed, I think patients are pretty receptive."*

-Pro 203

Of the providers we surveyed, 54% in both the intervention-exposed and unexposed groups reported that they had approached a patient about discontinuing an ICS. Of these, two providers in the unexposed group (15%) and one provider in the exposed group (8%) who proposed discontinuation of an ICS, reported that they were unsuccessful in discontinuing the ICS.

## Discussion

In a provider-randomized quality-improvement project, providers unexposed to the intervention of discontinuing a chronic medication anticipated concerns that were then reported and reflected by providers who were exposed to the intervention. Because the unexposed providers had not deprescribed an ICS in response to the intervention, their concerns of patient resistance or acceptance to the change in medication is framed as anticipatory. With the exception of lack of resistance to medication change, themes from patient interviews were consistent with interviews and survey findings from providers.

Congruent with previously published qualitative data from providers on hypothetical medication discontinuation, the unexposed group of providers anticipated that their patient would resist the discontinuation of an ICS because it seems counterintuitive to stop something that patients perceived as beneficial, patient's fear of worsening symptoms, or patient deference or loyalty to another provider who initiated the ICS prescription [9, 41]. These same reasonings for patient resistance were reported by providers who did undergo the intervention in both interviews and post-intervention surveys.

Based on the current literature [11, 41], we anticipated that some patients might exhibit psychological reactance, meaning specifically that they would direct anger and/or mistrust at the provider or the VA, with providers potentially being concerned about harming patient trust and the therapeutic relationship. We failed to find evidence of this in patient and provider interviews. Providers exposed to the intervention did report being asked by their patients why they needed to change a medication that was working fine. Though this could be perceived as an example of counter-arguing, we felt that based on the context, this finding was an example of patients' inquisitiveness rather than aggressive refusal to accept ICS discontinuation. Patients that we interviewed did not actively oppose the change, express anger or counter-argue (i.e., psychological reactance).

Rather, we found what appeared to be resignation to the medication change suggesting that perhaps patients don't feel strongly (or strongly empowered) about chronic inhalers. This resignation to the medication change aligned with previously reported findings. In their findings from interviews with older adults and caregivers, Weir *et al* report that patients tend to defer decisions regarding medications to their doctors. Similarly in a synthesis of qualitative studies on deprescribing, Bokhof and Junius-Walker describe that patients tend to "trustfully hand over" the responsibility of medication regimens to their physicians [41, 42]. Our findings of patient assent to deprescribing of ICS was in keeping with findings from Tannenbaum et al. where 62% of older adults who received a direct-to-consumer education intervention to reduce use of benzodiazepine, approached their physician or pharmacist [14, 15].

The absence of anger and skepticism on the part of patients does not mean the providers' concerns of patient resistance were ill-founded. A minority of patients do appear to express reluctance, primarily in the form of fear or concern over negative consequences, describing the ICS as a "security blanket". A small proportion of surveyed providers who proposed ICS discontinuation were unsuccessful (11.5%) at deprescribing it but did not provide further information on why. This may indicate patient resistance that we were not able to capture in our interviews. Future research could explore the characteristics of these patients who resist deprescribing, their disease processes, and their reasoning for rejecting the discontinuation of

a potentially harmful medication. Even if relatively rare, interactions with patients who experience distress over having a medication taken away might be highly salient when providers consider de-prescribing in the future.

Similar to the theme of patient resistance, we observed that unexposed providers anticipated and intervention-exposed providers reported patient acceptance to the medication change, particularly referencing the use of certain patient centered strategies that reflected previously published physician perspectives on hypothetical reduction of polypharmacy [41]. Providers reported proposing discontinuing the medication on a trial basis, allowing the patient to return using the ICS if there are negative consequences, or anticipating broaching the topic not as a decision by the PCP but a conversation with the patient. Furthermore, PCPs described initiating a discussion but being prepared to let it go if patients were unwilling and return to in the future. These strategies indicated providers respect for patient autonomy and may be one reason we did not find psychological reactance. Along the same lines, patients and providers reported that a trusting patient and provider relationship was key to accepting medication change echoing the current literature [41–44].

While patient centered care and strategies were a way to gain treatment alignment with patient, some of these providers acknowledged resignation to patient resistance that may be re-in forced by systemic barriers to deprescribing, such as having multiple prescribers from different medical organizations. These finding describes the wide range of patient reactions when discontinuing potentially harmful care depending on the context and urgency of the change proposed.

Patients that we interviewed reported generally deferring medication changes to their provider. Even when they expressed certain fears at not having the ICS, they were still amenable to a trial off it. In addition, 76% of all the providers who proposed discontinuing the ICS found that their patients were receptive. Despite this, interviews with providers suggest that anticipated patient reaction remains a major barrier. De-implementing interventions could target anticipated patient resistance for improved acceptance. We hypothesize that once providers begin deprescribing, they may develop more self-efficacy, especially by building strategies to counter potential resistance.

## Limitations and strengths

Our findings have a few noteworthy limitations. There could be response bias, and we do not know if the patients or providers who did not participate in interviews or surveys, and who received the DISCUSS COPD intervention, had similar experiences. The survey item asking respondents how patients responded if the respondent had proposed discontinuing an ICS did not orient the respondent to a specific patient (e.g., 'for the last patient whose ICS you proposed discontinuing. . .'), which may have introduced variation in how respondents interpreted the item. ICS may not engender a strong sense of need among patients compared to an addictive medication such as an opioid or a benzodiazepine. Patients taken off ICS are often prescribed an easy to take LAMA/LABA combination inhaler, so they may experience deprescription simply as a substitution rather than having a medication taken away. Finally, Veterans enrolled in VA healthcare system may be different from the general population, in that they have differing values and perspectives on healthcare and the perceived degree of autonomy in healthcare decisions when compared to the general population.

Another limitation is the low number of participants and the potential for selection bias. We were only able to conduct 6 interviews with intervention-exposed providers and 9 interviews with patients. Providers and patients who were willing to talk with us may have been more likely to have had a positive experience from deprescribing. While these numbers are

small, we feel that we did fulfill the objectives of our study and achieved saturation of themes in our interviews with unexposed providers and patients [45]. We reported a survey response rate of 35% with similar dangers to sampling bias. Finally, we were not able to interview provider-patient dyads, so are unable to compare provider and patient perspectives about the same instance of deprescribing.

This study had several important strengths as well, including that it was conducted as part of a prospective, provider-randomized quality-improvement project, in which the medications were deprescribed. Therefore, we have higher confidence that patient reactions were in response to the experience of deprescribing, and not the inverse (i.e., high-receptivity to deprescribing in patients leading to the ICS being deprescribed). There has been limited research on patient response to de-implementation of low-value care, and most of that work has been hypothetical, asking patients to respond to the idea of de-implementation, not the experience of it. We obtained both qualitative and quantitative data, and interviewed both PCPs and patients, giving us a fuller picture of how patients reacted, both from their perspective and from the perspective of their providers.

## Conclusions

Deprescribing medications can have multiple challenges. In implementing a systematic de-implantation of a potentially harmful medication, we found that while providers anticipated resistance from patients, that providers were largely versatile in responding to patient fears and concerns by coming up with patient centered approaches. We found that despite our intervention, several providers did not bring up deprescribing with their patients, potentially fearing patient resistance. When designing interventions to deprescribe medications, focusing on provider anticipation of resistance and providing tools to counter the potential resistance could help facilitate the act of deprescribing.

## Supporting information

**S1 File. Unexposed provider interview guide.**
(DOCX)

**S2 File. Intervention-exposed provider interview guide.**
(DOCX)

**S3 File. Intervention-exposed patient interview guide.**
(DOCX)

**S4 File. COREQ: 32-item checklist.**
(DOCX)

**S1 Table. Interview participants.**
(DOCX)

**S2 Table. Matrix of quotes relevant to patient reaction.**
(DOCX)

## Acknowledgments

We are grateful to the Veterans and providers who participated in this work in order to improve the quality of care we deliver. We thank our research assistant Scott Wanner who helped with data collection.

**Disclaimer**: The views expressed in this article are those of the authors and do not necessarily reflect the position or policy of the Department of Veterans Affairs, the United States Government, and the affiliated institutions.

## Author Contributions

**Conceptualization:** Toral J. Parikh, Christian D. Helfrich.

**Data curation:** Krysttel C. Stryczek, Chris Gillespie, Barbara Majerczyk.

**Formal analysis:** Toral J. Parikh, Krysttel C. Stryczek, Chris Gillespie, George G. Sayre, Laura Feemster, Seppo T. Rinne, Renda Soylemez Wiener, Christian D. Helfrich.

**Funding acquisition:** David H. Au, Christian D. Helfrich.

**Investigation:** Toral J. Parikh, Krysttel C. Stryczek, Chris Gillespie, Edmunds Udris, Barbara Majerczyk.

**Project administration:** Edmunds Udris, Barbara Majerczyk.

**Software:** Krysttel C. Stryczek, Chris Gillespie.

**Supervision:** George G. Sayre, David H. Au, Christian D. Helfrich.

**Writing – original draft:** Toral J. Parikh.

**Writing – review & editing:** Toral J. Parikh, Krysttel C. Stryczek, Chris Gillespie, George G. Sayre, Laura Feemster, Edmunds Udris, Barbara Majerczyk, Seppo T. Rinne, Renda Soylemez Wiener, David H. Au, Christian D. Helfrich.

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
