## [Decision Letter · Decision Letter 0]

15 Apr 2020

PONE-D-20-04291

Patient reaction when deprescribing a chronic medication: findings from a quality improvement project

PLOS ONE

Dear Dr. Parikh,

Thank you for submitting your manuscript to PLOS ONE. After careful consideration, we feel that it has merit but does not fully meet PLOS ONE’s publication criteria as it currently stands. Therefore, we invite you to submit a revised version of the manuscript that addresses the points raised during the review process.

This paper has been reviewed by two experts in the field who confirm that the work is important and adds to the literature.  However, Reviewer 1 makes a number of comments regarding the methods and analysis, as well as how the work is reported; please ensure the work is reported to the COREQ checklist, as suggested by the reviewer.

We would appreciate receiving your revised manuscript by May 29 2020 11:59PM. To enhance the reproducibility of your results, we recommend that if applicable you deposit your laboratory protocols in protocols.io, where a protocol can be assigned its own identifier (DOI) such that it can be cited independently in the future. For instructions see: http://journals.plos.org/plosone/s/submission-guidelines#loc-laboratory-protocols

We look forward to receiving your revised manuscript.

Kind regards,

Adam Todd, PhD

Academic Editor

PLOS ONE

Reviewers' comments:

Reviewer's Responses to Questions

**Comments to the Author**

1. Is the manuscript technically sound, and do the data support the conclusions?

Reviewer #1: Yes

Reviewer #2: Yes

2. Has the statistical analysis been performed appropriately and rigorously? 

Reviewer #1: N/A

Reviewer #2: N/A

3. Have the authors made all data underlying the findings in their manuscript fully available?

Reviewer #1: No

Reviewer #2: Yes

4. Is the manuscript presented in an intelligible fashion and written in standard English?

Reviewer #1: Yes

Reviewer #2: Yes

5. Review Comments to the Author

Reviewer #1: This is a very interesting article and the results contribute to the field. There are areas of the methods that require greater detail and I have a number of other comments which I think will improve the clarity and usefulness of the article to readers.

Title – Suggesting mentioning ICS in the title to increase the descriptiveness – potentially also with the detail that it was guideline disconcordant use of ICS.

Abstract – Suggest adding a summary of the results from the patient interviews (currently only mentions that they accepted the recommendation).

Introduction

The reference to Reeve’s focus group study are not completely accurate – the barriers are appropriately described, but the enablers found in the study are not mentioned. In the text this study is contrasted with another which reported openness to deprescribing which makes it seem like no willingness was reported in these focus groups which is not true.

Additionally, you may want to include reference to the studies using the Patients’ Attitudes Towards Deprecribing (PATD/rPATD) questionnaire regarding patient willingness to have a medication deprescribed. Including in some groups on specific drugs

e.g.

Edelman, M, et al. "Patients’ Attitudes Towards Deprescribing Alpha-Blockers and Their Willingness to Participate in a Discontinuation Trial." Drugs & aging 36.12 (2019): 1133-1139.

Reeve, E, et al. "Assessment of attitudes toward deprescribing in older Medicare beneficiaries in the United States." JAMA internal medicine 178.12 (2018): 1673-1680.

Methods

It’s not clear why PCPs who had not been in the intervention were recruited for this study. Their perception of how a patient might react seems less important compared to PCPs who have recently conducted deprescribing of a ICS with a patient.

It appears that the findings reported here are a sub-result of the interviews with PCPs – (De-implementing Inhaled Corticosteroids to Improve Care and Safety in COPD Treatment: Primary Care Providers' Perspectives) – this needs to be made clear in this article. This has important implications for how the analysis was conducted – e.g. are these results/themes created simultaneously with the others and then just reported separately here?

There are many recommended elements for reporting qualitative research missing in this article – for example, what was the relationship of the interviewer to the participants, how many people were involved in the coding, data saturation etc – see COREQ or other similar tools. Also, saturation is mentioned in the discussion but not in methods or results?

Results

It is not clear which themes were deductively versus inductively created?

Patient resistance – while all the pts interviewed agreed to stop – authors report one person who had fear about stopping – it would be interesting to know if there was discussion of their decision making process, weighing up of pros and cons – why did the person who had fears not debate the decision?

“The post-intervention surveys confirmed that generally patients were receptive to the conversation regarding stopping an ICS or substituting with another drug.” – please add the detail that this was according to providers not patients.

Very little results from patients seem to be presented especially in comparison to the interview guide in the supplementary material – are the authors planning on publishing other aspects from the interviews elsewhere. E.g. experience of the conversation, decision making process?

Discussion

“Providers reported proposing discontinuing the medication on a trial basis, allowing the patient to return using the ICS if there are negative consequences, or anticipating broaching the topic not as a decision by the PCP but a conversation with the patient.” – this note of trial basis and restarting is not stated anywhere in the results?

“Further support in de-implementing interventions could target these instances to make the intervention more successful.” – Given that 46% of exposed providers never approached the topic what does that mean in the context of the intervention in the parent study – some discussion of what this intervention was would help put these results in context – i.e. a brief description of the DISCUSS COPD program. This is highlighted in the conclusion but not really mentioned in results or discussion.

The authors note in the introduction that 2 previous studies have conducted qual study in response to a deprescribing intervention – I would have appreciated a specific discussion of how the results of this study compared with these others - 13. Martin P, Tannenbaum C. A realist evaluation of patients’ decisions to deprescribe in the 543 EMPOWER trial. BMJ Open. 2017 May;7(4):e015959. 14. Walsh K, Kwan D, Marr P, Papoushek C, Lyon WK. Deprescribing in a family health team: a study of chronic proton pump inhibitor use. J Prim Health Care. 2016;8(2):164.

Limitations – Suggest adding that there were no NPs and no PAs in intervention exposed participant groups.

Conclusion –

Recommend reducing the length and focusing on the actual conclusions that directly result from this study (the first paragraph of the conclusion repeats background and aims).

Reviewer #2: Thank you for the opportunity to review your manuscript, “Patient reaction when deprescribing a chronic medication: Findings from a quality improvement project.” This manuscript discusses a mixed methods analysis of a quality improvement project focused on discontinuation of inappropriate inhaled corticosteroids at two VA health care systems. This manuscript was well written and adds to the literature by providing patient and provider perspectives about a specific deprescribing intervention, as opposed to responding to hypothetical scenarios.

Here are a few minor suggestions to add clarity to the manuscript:

• Participants and recruitment: Recommend clarify how you purposively sampled primary care providers and patients.

• Table 3

o Were the questions about prescribing or discontinuing ICS specific to COPD?

o It is difficult to interpret the results in the “if yes” section. For example, were they thinking back for the last 6 months? How did they report data if the worked with multiple patients on potentially stopping ICS?

6. PLOS authors have the option to publish the peer review history of their article (what does this mean?). If published, this will include your full peer review and any attached files.

Reviewer #1: No

Reviewer #2: No

---

## [Author Response · Author response to Decision Letter 0]

29 May 2020

Reviewer #1: This is a very interesting article and the results contribute to the field. There are areas of the methods that require greater detail and I have a number of other comments which I think will improve the clarity and usefulness of the article to readers.

>>Thank you for your feedback and the opportunity to respond to it.

Title – Suggesting mentioning ICS in the title to increase the descriptiveness – potentially also with the detail that it was guideline disconcordant use of ICS.

>>Thank you for this feedback. We have changed the title to more accurately reflect the focus of our analysis. “Provider anticipation and experience of patient reaction when deprescribing guideline discordant inhaled corticosteroids”

Abstract – Suggest adding a summary of the results from the patient interviews (currently only mentions that they accepted the recommendation).

>>We have followed your suggestion.

Introduction

The reference to Reeve’s focus group study are not completely accurate – the barriers are appropriately described, but the enablers found in the study are not mentioned. In the text this study is contrasted with another which reported openness to deprescribing which makes it seem like no willingness was reported in these focus groups which is not true.

Additionally, you may want to include reference to the studies using the Patients’ Attitudes Towards Deprecribing (PATD/rPATD) questionnaire regarding patient willingness to have a medication deprescribed. Including in some groups on specific drugs

e.g.Edelman, M, et al. "Patients’ Attitudes Towards Deprescribing Alpha-Blockers and Their Willingness to Participate in a Discontinuation Trial." Drugs & aging 36.12 (2019): 1133-1139.

Reeve, E, et al. "Assessment of attitudes toward deprescribing in older Medicare beneficiaries in the United States." JAMA internal medicine 178.12 (2018): 1673-1680.

>>Thank you for this suggestion. We have included both of these studies in the introduction (lines 112-122). However, we would like to emphasize that the Reeve et. al. asks for general views on medication discontinuation while the Edelman et al. article elicits patient response to a hypothetical question if the provider proposed discontinuation. However patient reaction (as reported by patients and anticipated/experienced by providers) has not been studied in the specific context of a deprescribing intervention.

Methods

It’s not clear why PCPs who had not been in the intervention were recruited for this study. Their perception of how a patient might react seems less important compared to PCPs who have recently conducted deprescribing of a ICS with a patient.

>>Thank you for the opportunity to clarify this important point. The current analysis was done as a continuation of previously published work (Stryczek et. al.). We have referenced it in the manuscript several times (https://link.springer.com/article/10.1007/s11606-019-05193-2). This body of work includes providers unexposed to the intervention to understand their pre-existing attitudes and perspectives, as well as the perspectives of intervention exposed providers. The unexposed providers’ perspectives were equally informative for understanding patient reaction as a barrier to de-prescribing as were the intervention providers. The same interview guide from Stryczek et al. was used to assess unexposed providers views. 

It appears that the findings reported here are a sub-result of the interviews with PCPs – (De-implementing Inhaled Corticosteroids to Improve Care and Safety in COPD Treatment: Primary Care Providers' Perspectives) – this needs to be made clear in this article. This has important implications for how the analysis was conducted – e.g. are these results/themes created simultaneously with the others and then just reported separately here?

>>Yes, the current analysis was done as a continuation of previously published work (Stryczek et. al. as referenced in the above comment and added to the manuscript) (lines 157-159). However, provider responses regarding anticipation of patient reaction was an emergent, but preliminary finding identified in the unexposed providers. At the time of the first publication, there was not sufficient data to report these findings until data collection with intervention-exposed providers and patient interviews was completed for full analysis. 

There are many recommended elements for reporting qualitative research missing in this article – for example, what was the relationship of the interviewer to the participants, how many people were involved in the coding, data saturation etc – see COREQ or other similar tools. Also, saturation is mentioned in the discussion but not in methods or results?

>>We referred to the COREQ tool during the preparation of this manuscript and reported the applicable elements where appropriate. 

>>Team members responsible for qualitative data collection and coding were specified on the cover page and additional details were published in Stryczek et al.

>>Regarding qualitative team member relationships with participants: some participants interviewed were clinician-researchers that have worked with members of the qualitative team (2 of 3 interviewers) in the past on other research projects. However, the interviewers are not clinicians, and interviews were completed with clinician-researchers within the scope of their clinician role. The interviewers were not working with participants on any projects at the time that the interviews were completed. No participants were involved in this project in any way.

>>As far as recruitment for interviews, data collection for providers continued until no further participants volunteered, and data collection for patients continued until data saturation was reached. We have revised the methods to described this more clearly how saturation was determined.

Results

It is not clear which themes were deductively versus inductively created?

>>The topic and themes presented in this manuscript were emergent findings. Meaning that we had noted preliminary findings in intervention unexposed providers anticipating patient reaction. We then looked at interviews from intervention exposed providers and patients to analyze if patient reaction was a prevalent topic in all three groups of interviews.

Patient resistance – while all the pts interviewed agreed to stop – authors report one person who had fear about stopping – it would be interesting to know if there was discussion of their decision making process, weighing up of pros and cons – why did the person who had fears not debate the decision?

>>This was a very interesting point. The patients that we interviewed did not elaborate on their decision-making process, even with appropriately grounded prompts and probing. During the interviews, patients reported deferring to their providers. 

“The post-intervention surveys confirmed that generally patients were receptive to the conversation regarding stopping an ICS or substituting with another drug.” – please add the detail that this was according to providers not patients.

>>We clarified this as suggested.

Very little results from patients seem to be presented especially in comparison to the interview guide in the supplementary material – are the authors planning on publishing other aspects from the interviews elsewhere. E.g. experience of the conversation, decision making process?

>>We state the goals of this particular analysis on Page 6 last paragraph (lines 151-159) starting with, “Our goal….” We included all the pertinent quotes relating to this analysis from provider and patient interviews. Despite the comprehensive interview guide, patient responses were brief and to the point without much elaboration. As stated above, patients did not elaborate on decision making process despite probing from interviewers. At this time, the authors are unclear if further data from the patient interviews will sufficiently add to future manuscripts. 

Discussion

“Providers reported proposing discontinuing the medication on a trial basis, allowing the patient to return using the ICS if there are negative consequences, or anticipating broaching the topic not as a decision by the PCP but a conversation with the patient.” – this note of trial basis and restarting is not stated anywhere in the results?

>>Thank you for this observation. We have amended the results section so that our discussion elaborates on the results. We added a specific quote where one provider mentions discontinuing an ICS on a trial basis. (lines 382-385)

“Further support in de-implementing interventions could target these instances to make the intervention more successful.” – Given that 46% of exposed providers never approached the topic what does that mean in the context of the intervention in the parent study – some discussion of what this intervention was would help put these results in context – i.e. a brief description of the DISCUSS COPD program. This is highlighted in the conclusion but not really mentioned in results or discussion.

>>Thank you for pointing this out. Our intervention was a proactive e-consult generated by the pulmonary team targeted to primary care providers for recommendations to discontinue ICS when appropriate or other relevant recommendations. This is now explained in detail under Materials and Methods (lines 164-170). While preliminary data suggests that the e-consults were widely accepted, we cannot yet put the survey data into the context of how successful the e-consult intervention was. Our survey was not able to provide further context for this finding.

The authors note in the introduction that 2 previous studies have conducted qual study in response to a deprescribing intervention – I would have appreciated a specific discussion of how the results of this study compared with these others - 13. Martin P, Tannenbaum C. A realist evaluation of patients’ decisions to deprescribe in the 543 EMPOWER trial. BMJ Open. 2017 May;7(4):e015959. 14. Walsh K, Kwan D, Marr P, Papoushek C, Lyon WK. Deprescribing in a family health team: a study of chronic proton pump inhibitor use. J Prim Health Care. 2016;8(2):164.

>>Thank you for this close observation. Upon review, the Walsh et. al. study reported findings from post-intervention provider surveys on the user feasibility and acceptability of the intervention rather than provider and patient experience as a result of the intervention. We have removed it as a reference. The Martin and Tannenbaum (2018) study has now been referenced in both the introduction and discussion (lines 448-451).

Limitations – Suggest adding that there were no NPs and no PAs in intervention exposed participant groups.

>>Thank you for pointing this out. It is correct that we did not have PAs in our sample, but we did have NPs enrolled in our intervention. We were able to recruit two NPs for unexposed provider interviews but none for the 6 interviews we did with intervention-exposed providers. We believe that this was a product of small sample size, which we address in our limitations

Conclusion –

Recommend reducing the length and focusing on the actual conclusions that directly result from this study (the first paragraph of the conclusion repeats background and aims).

>>Thank you for this suggestion. We have attempted to reduce the length and focus on the actual conclusions; we removed the first summarizing paragraph from the conclusion.

Reviewer #2: Thank you for the opportunity to review your manuscript, “Patient reaction when deprescribing a chronic medication: Findings from a quality improvement project.” This manuscript discusses a mixed methods analysis of a quality improvement project focused on discontinuation of inappropriate inhaled corticosteroids at two VA health care systems. This manuscript was well written and adds to the literature by providing patient and provider perspectives about a specific deprescribing intervention, as opposed to responding to hypothetical scenarios.

>>Thank you for reviewing the paper and opportunity to revise it.

Here are a few minor suggestions to add clarity to the manuscript:

Participants and recruitment: Recommend clarify how you purposively sampled primary care providers and patients.

>>Thank you for this recommendation. We have included a more detailed description in the methods (lines 184-191, 193-202). 

>>“One cohort of eligible providers were recruited for qualitative interviews prior to delivery of the DISCUSS COPD deprescribing intervention. These providers were not exposed to the intervention (unexposed). The second cohort included eligible providers exposed to the DISCUSS COPD deprescribing intervention (intervention-exposed). These providers were recruited for qualitative interviews after they received at least three DISCUSS COPD e-consults. Qualitative data collection continued until there were no new responses from PCPs to participate. Both groups of PCPs (unexposed and intervention-exposed) were recruited for quantitative surveys after the intervention was implemented.” (lines 184-191)

>>“Patients of intervention-exposed PCPs were also purposively sampled for qualitative interviews if they were identified by the DISCUSS COPD Pulmonology team for the deprescribing intervention and the intervention was documented in the provider-patient encounter.” (lines 195-195)

Table 3

Were the questions about prescribing or discontinuing ICS specific to COPD?

>>The survey questions about prescribing or discontinuing ICS were specific to mild-to-moderate COPD. We have now included specific wording from the survey for increased clarity and transparency (lines 244-255). 

It is difficult to interpret the results in the “if yes” section. For example, were they thinking back for the last 6 months? How did they report data if the worked with multiple patients on potentially stopping ICS?

>>The authors also had similar concerns. It has been noted in the limitations section: “The survey item asking respondents how patients responded if the respondent had proposed discontinuing an ICS did not orient the respondent to a specific patient (e.g., ‘for the last patient whose ICS you proposed discontinuing…’), which may have introduced variation in how respondents interpreted the item.” (lines 493-496)

---

## [Decision Letter · Decision Letter 1]

16 Jun 2020

PONE-D-20-04291R1

Provider anticipation and experience of patient reaction when deprescribing guideline discordant inhaled corticosteroids

PLOS ONE

Dear Dr. Parikh,

Thank you for submitting your manuscript to PLOS ONE. After careful consideration, we feel that it has merit but does not fully meet PLOS ONE’s publication criteria as it currently stands. Therefore, we invite you to submit a revised version of the manuscript that addresses the points raised during the review process.

Please ensure the work is reported to the COREQ checklist, and please make it clear in the paper as a standalone document.  As Reviewer 1 points out, this is important for future research in this area.  

We look forward to receiving your revised manuscript.

Kind regards,

Adam Todd, PhD

Academic Editor

PLOS ONE

Reviewers' comments:

Reviewer's Responses to Questions

**Comments to the Author**

1. If the authors have adequately addressed your comments raised in a previous round of review and you feel that this manuscript is now acceptable for publication, you may indicate that here to bypass the “Comments to the Author” section, enter your conflict of interest statement in the “Confidential to Editor” section, and submit your "Accept" recommendation.

Reviewer #1: (No Response)

Reviewer #2: All comments have been addressed

2. Is the manuscript technically sound, and do the data support the conclusions?

Reviewer #1: Yes

Reviewer #2: (No Response)

3. Has the statistical analysis been performed appropriately and rigorously? 

Reviewer #1: Yes

Reviewer #2: (No Response)

4. Have the authors made all data underlying the findings in their manuscript fully available?

Reviewer #1: Yes

Reviewer #2: (No Response)

5. Is the manuscript presented in an intelligible fashion and written in standard English?

Reviewer #1: Yes

Reviewer #2: (No Response)

6. Review Comments to the Author

Reviewer #1: I appreciate the important work reported in this article and the responses from the authors. I just have two outstanding comments that while the authors responded to in the 'responses to reviewers' section, the changes made to the manuscript didn't fully provide the clarification necessary.

1. Lack of clarity of inductive vs deductive themes and how this work correlates to the previously published article by Stryczek et al.

The methods reported in line 229-237 mention inductive and deductive content analysis – but it is not clear whether the theme presented in this article was a result of inductive vs deductive. I think it would help to provide some timeline/context which was somewhat covered in the response document (but no changes were made to the manuscript). E.g. to say the unexposed/pre-intervention interviews were conducted with deductive and inductive approach. And one very interesting emergent (inductive) theme from these was this theme of patient reaction. Therefore in the exposed provider and patient interviews these theme was further explored – so this manuscript presents the results of all 3 interviews in relation to this theme. i.e. did the theme of patient reaction come from the previous lit and then the subthemes were all inductive, or was the theme and sub-themes all from previous lit. Or was the theme and subthemes all inductive from the first lot of interviews (unexposed) and then the 2 second lots of interviews coded to these themes and sub-themes with or without further refinement?

This article still needs to be standalone without the readers having to read the previous article.

2. Items from the COREQ checklist not covered.

The authors provided information in their responses which have not been added to the article. The authors say "We referred to the COREQ tool during the preparation of this manuscript and reported the applicable elements where appropriate." But key info such as whether there was duplicate coding, relationships etc is not in this article. Again, this article still needs to be standalone without the readers having to read the previous article. I strongly suggest adding the COREQ checklist with appropriate information as a supplementary file to this manuscript. While this may not seem essential to the presentation of the results, having this information clearly available is essential for future research in this field (such as conducting systematic reviews).

Reviewer #2: Thank you for the opportunity to re-review the manuscript, “Provider anticipation and experience of patient reaction when deprescribing guideline discordant inhaled corticosteroids.” Thank you for adding additional information throughout the manuscript. It is easy to read and includes pertinent information. Below are two minor suggestions:

Title:

- The title is more descriptive but is now quite long. Consider whether it would be possible to summarize it further (e.g., Provider and patient perceptions when deprescribing guideline discordant inhaled corticosteroids.”)

Qualitative interviews:

- Line 227: Please clarify if “perspectives on medical overuse” was explored and not “perspectives on medication overuse.” Recommend adding “and.”

7. PLOS authors have the option to publish the peer review history of their article (what does this mean?). If published, this will include your full peer review and any attached files.

Reviewer #1: No

Reviewer #2: No

---

## [Author Response · Author response to Decision Letter 1]

31 Jul 2020

From Editor:

Please ensure the work is reported to the COREQ checklist, and please make it clear in the paper as a standalone document. As Reviewer 1 points out, this is important for future research in this area.

--Thank you for the opportunity to address reviewer concerns and comments. We have included a COREQ checklist and done our best to make this manuscript a standalone paper. 

From Reviewer 1:

I appreciate the important work reported in this article and the responses from the authors. I just have two outstanding comments that while the authors responded to in the 'responses to reviewers' section, the changes made to the manuscript didn't fully provide the clarification necessary.

--Thank you for allowing us to respond to your concerns and further clarifying our methods. 

Lack of clarity of inductive vs deductive themes and how this work correlates to the previously published article by Stryczek et al.

The methods reported in line 229-237 mention inductive and deductive content analysis – but it is not clear whether the theme presented in this article was a result of inductive vs deductive. I think it would help to provide some timeline/context which was somewhat covered in the response document (but no changes were made to the manuscript). E.g. to say the unexposed/pre-intervention interviews were conducted with deductive and inductive approach. And one very interesting emergent (inductive) theme from these was this theme of patient reaction. Therefore in the exposed provider and patient interviews these theme was further explored – so this manuscript presents the results of all 3 interviews in relation to this theme. i.e. did the theme of patient reaction come from the previous lit and then the subthemes were all inductive, or was the theme and sub-themes all from previous lit. Or was the theme and subthemes all inductive from the first lot of interviews (unexposed) and then the 2 second lots of interviews coded to these themes and sub-themes with or without further refinement?

This article still needs to be standalone without the readers having to read the previous article.

-- Thank you for allowing us to clarify this important point. We have added the following:

--“One inductive theme that emerged amongst barriers reported by unexposed providers was anticipation of patient reaction when discussing the discontinuing or substituting the ICS. As stated earlier, the interview guides for unexposed providers, intervention exposed providers, and patients were iteratively refined to allow further elaboration on patient reaction.” (Lines 240-243)

--“This analysis builds on previously published work on unexposed providers [10]. Patient reaction, however, was an emergent theme for which data collection from intervention-exposed providers and patient interviews had continued well after the initial publication. The current findings further explore this emergent theme of patient reaction and includes analysis of data from intervention-exposed providers and patient interviews. Given the depth and subtle but important variations of this theme, we present a detailed analysis after synthesizing and incorporating both qualitative and quantitative data.” (Lines 276-281) 

Items from the COREQ checklist not covered.

The authors provided information in their responses which have not been added to the article. The authors say "We referred to the COREQ tool during the preparation of this manuscript and reported the applicable elements where appropriate." But key info such as whether there was duplicate coding, relationships etc is not in this article. Again, this article still needs to be standalone without the readers having to read the previous article. I strongly suggest adding the COREQ checklist with appropriate information as a supplementary file to this manuscript. While this may not seem essential to the presentation of the results, having this information clearly available is essential for future research in this field (such as conducting systematic reviews).

--We have included a completed COREQ checklist and updated necessary information in the methods to make this a stand-alone manuscript as recommended. Thank you. 

--“Interviews lasted approximately 20-30 minutes, were audio-recorded and transcribed verbatim to facilitate analyses. Data collection was completed by authors CG, GGS, and KCS, and research assistant SW. Qualitative software (ATLAS.ti 7; Scientific Software Development GmbH, Berlin, Germany) was used for data management and coding. Coding and analyses were completed by team members (CG, CDH, GGS, KCS, and TJP) involved in data collection and members of the larger qualitative team trained in medicine and health services research.” (Lines 210-215)

From Reviewer 2:

Thank you for the opportunity to re-review the manuscript, “Provider anticipation and experience of patient reaction when deprescribing guideline discordant inhaled corticosteroids.” Thank you for adding additional information throughout the manuscript. It is easy to read and includes pertinent information. Below are two minor suggestions:

--Thank you for the suggestions. 

The title is more descriptive but is now quite long. Consider whether it would be possible to summarize it further (e.g., Provider and patient perceptions when deprescribing guideline discordant inhaled corticosteroids.”)

--Thank you for this recommendation. Unfortunately, truncating the title implies that we are going to address all patient and provider perceptions regarding deprescribing of ICS. That has been done in our previous publication. In this paper, we did a deeper and focused analysis on patient reaction as it was anticipated and experienced. We want our title to reflect that limited focus to avoid misleading our readers as we had with our original title. 

Line 227: Please clarify if “perspectives on medical overuse” was explored and not “perspectives on medication overuse.” Recommend adding “and.”

--We have made the following revisions: “changes in their inhaler prescription, and perspectives on medication overuse” (Lines 231-232)

---

## [Editor Report · Decision Letter 2]

19 Aug 2020

Provider anticipation and experience of patient reaction when deprescribing guideline discordant inhaled corticosteroids

PONE-D-20-04291R2

Dear Dr. Parikh,

We’re pleased to inform you that your manuscript has been judged scientifically suitable for publication and will be formally accepted for publication once it meets all outstanding technical requirements.

Kind regards,

Adam Todd, PhD

Academic Editor

PLOS ONE
---

## [Editor Report · Acceptance letter]

8 Sep 2020

PONE-D-20-04291R2 

Provider anticipation and experience of patient reaction when deprescribing guideline discordant inhaled corticosteroids 

Dear Dr. Parikh:

I'm pleased to inform you that your manuscript has been deemed suitable for publication in PLOS ONE. Congratulations! Your manuscript is now with our production department. 

Kind regards, 

on behalf of

Dr. Adam Todd 

Academic Editor

PLOS ONE